# Monitoring of Beech Glued Laminated Timber and Delamination Resistance of Beech Finger-Joints in Varying Ambient Climates

**Hannes Stolze** [1,*] , **Mathias Schuh** [2], **Sebastian Kegel** [1], **Connor Fürkötter-Ziegenbein** [1], **Christian Brischke** [1] and **Holger Militz** [1]

1. Wood Biology and Wood Products, Faculty of Forest Sciences and Forest Ecology, University of Goettingen, Buesgenweg 4, 37077 Goettingen, Germany; sebastian.kegel@stud.uni-goettingen.de (S.K.); cfuerkoetterziege@stud.uni-goettingen.de (C.F.-Z.); christian.brischke@uni-goettingen.de (C.B.); holger.militz@uni-goettingen.de (H.M.)
2. Holzforschung München, Chair of Wood Science, Technical University of Munich, Winzererstr. 45, 80797 Munich, Germany; schuh@hfm.tum.de
* Correspondence: hannes.stolze@uni-goettingen.de; Tel.: +49-551-39-33562

**Abstract:** In this study, varying ambient climates were simulated in a test building by changing temperature and relative humidity. Beech glued laminated timber (glulam, *Fagus sylvatica*, L.) was freshly installed in the test building and monitoring of the change in wood moisture content of the glulam resulting from the variations in climate was carried out. Subsequently, finger-jointed beech specimens were exposed to the variations in relative humidity measured in the course of the monitoring experiment on a laboratory scale, and thus an alternating climate regime was derived from the conditions in the test building. Its influence on the delamination of the finger-joints was evaluated. In addition, it was examined whether beech finger-joints using commercial adhesive systems fulfil the normative requirements for delamination resistance according to EN 301 (2018) and whether different bonding-wood moisture levels have an effect on the delamination of the finger-joints. In the context of the monitoring experiment, there was a clear moisture gradient in the beech glulam between the inner and near-surface wood. The applied adhesive systems showed almost the same delamination resistance after variation of relative humidity. The normative requirements were met by all PRF-bonded and by most PUR-bonded beech finger-joints with higher bonding wood moisture content.

**Keywords:** beech glulam; monitoring experiment; finger-joint bonding; delamination; adhesives

## 1. Introduction

Extreme climatic conditions often exist during the application phase of timber construction products [1] and especially during transport to or installation on the construction site. Timber construction products approved by the building authorities, regardless of whether they are made of softwood or hardwood, or are not bonded or bonded, must be able to withstand these conditions. The effects of applying an anhydrite screed with a high water content on the wood moisture content (MC) of beech glued laminated timber (beech glulam) were examined in this study under real conditions. Changes in MC and resulting gradients within a timber construction product can be associated with internal stresses, formation of cracks and, in the case of bonded products, with delamination of the adhesive joints [2]. For the above-mentioned reasons, high delamination resistance is of great relevance for bonded and load-bearing timber construction products and an important criterion for their reliability under extreme stress. The focus of the present study was on the delamination resistance of beech finger-joint bonding. Delamination testing, an accelerated ageing method, is used to test the resistance of bonding for later applications as part of quality and performance control. Since wood construction products are almost exclusively

made of coniferous wood and in particular of Norway spruce (*Picea abies*, L.) [3], the delamination test is designed for spruce [4,5]. Hardwood bondings such as those of beech, the most abundant hardwood in Central Europe, are also tested according to the standard designed for spruce, and currently there are only adapted requirements for the hardwood species oak (*Quercus* spp.) [4]. On beech surface bonding, references [6–8] showed that individual adhesive systems (partly without proof of applicability for hardwood) can fulfil the normative requirements for delamination resistance, but their results were, at the same time, dependent on a large number of parameters. Main influencing factors are the effects of the wood species and wood modifications, the type of adhesive and its processing, the quality of the bonding surface, and varying climatic conditions during production and in the use phase [9,10]. Various one-component polyurethane and two-component aminoplast adhesive systems used for finger-jointing beech and oak showed negative results compared to the normative requirements for delamination resistance [6]. In addition, hardwood finger-jointing has been less investigated compared to surface bonding. In order to establish construction products made of hardwood in timber construction, adaptations are necessary to improve the resistance of hardwood bonding, primarily for finger-joint bonding. The integration of adhesive systems and bonding technology is essential for this. At the same time, due to wood species-specific differences, it is necessary to give greater consideration to hardwoods in the standardisation of delamination testing and, if necessary, to specify conditions for individual hardwood species [2,6,11,12]. Finger-jointing, as an end-face joining technique, which is most frequently used in today's timber construction products, is particularly challenging for beech. This is due to the high strength of beech, which has to be transferred at the joints of a building component. With regard to the resistance of the bonding, the high moisture sorptivity and low dimensional stability of beech with resulting stresses represent a major challenge [13]. Accordingly, high demands are placed on the adhesive systems and properties such as the formation of elastic adhesive joints or mechanisms to reduce swelling and shrinking, which are needed to ensure a durable bond [14]. The aim of the present study was to show how the MC of beech glulam changes under a real construction application. In addition, the aim was to clarify whether beech finger-joints can fulfil the currently applied normative requirements for delamination resistance according to EN 301 (2018) [4]. Finally, it should be examined to what extent the conditions in a real construction application differ from the standard delamination tests and how beech finger-joints perform under respective conditions.

## 2. Materials and Methods

### 2.1. Specimens and Adhesives

The beech glulam used in this study was produced in the *KlimaKleb* project (16KN042025) as part of the Central Innovation Programme for medium-sized enterprises (ZIM). The glulam contained a 10-layer structure and had dimensions of $2450 \times 100 \times 215$ mm$^3$ (L × W × H). Finger-jointed delamination specimens with dimensions of $100 \times 175 \times 44$ mm$^3$ were produced from steamed beech logs (North Thuringia, Germany) with an initial thickness of 65 mm and a raw density of 0.7 g cm$^{-3}$ ($\pm 0.03$ g cm$^{-3}$). Three adhesive systems were selected as they are approved for load-bearing bonding of beech wood. They belong to the following adhesive types:

- One-component polyurethane (PUR).
- Two-component melamine-urea-formaldehyde (MUF)
- Two-component phenol-resorcinol-formaldehyde (PRF).

### 2.2. Monitoring Experiment of Beech Glued Laminated Timber in Varying Ambient Climates

In a test building (internal dimension: $3.96 \times 2.46$ m$^2$ (L × W)) of the company Bau-Fritz GmbH & Co. KG (Erkheim, Germany), beech glulam was installed as a joist ($2450 \times 100 \times 215$ mm$^3$) and placed on two beech glulam supports. The test stand was equipped with a 1.6 m wide window for ventilation. After the installation of the beech glulam and the heating and measuring equipment, anhydrite screed (Knauf KG,

flowing screed) with a high water content was poured into the test building for self-levelling. This was followed by a heating and drying phase according to a defined program, using underfloor heating and a mobile construction heater (Figure 1) as well as specified ventilation intervals, as follows:

- Day 1: Pouring the anhydrite screed
- Day 3: First ventilation after 48 h, afterwards 1–3 × daily ventilation (each 10–15 min, weekdays)
- Day 7: Start the heating program 7 days after pouring the anhydrite screed
- Day 7–10: Heat-up phase up to 25 °C
- Day 11–18: Increase the room temperature by 5 °C/day until 45 °C was reached
- Day 19–32: Maintain a constant room temperature of 45 °C (±5 °C) for 14 days

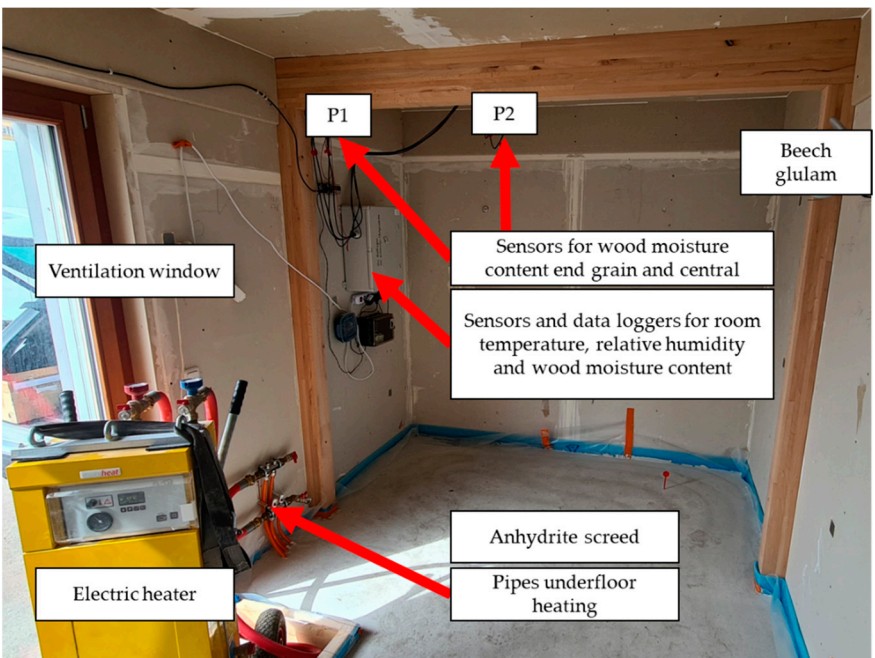

**Figure 1.** Monitoring experiment of beech glulam in the test bulding and positions of the measurement equipment.

During the test phase, temperature (T) and relative humidity (RH) were continuously recorded in the test building at 10-min intervals. A LOG-HC2-P1 data logger and HC2A-SH sensors (Rototronic Messgeräte GmbH, Ettlingen, Germany) were used for this purpose.

In addition, the MC of the beech glulam was measured 15–25 cm from the end grain (Figure 1, P1) and in the centre of the glulam (P2) via insulated electrodes at an effective measuring depth of 0–20 mm, 10–20 mm, 30–40 mm, and 40–50 mm, using a Gigamodul multi-channel moisture meter (Scanntronic, Mugrauer GmbH, Zorneding, Germany). In this way, the change in MC at different depths was recorded (Figure 2).

In this study, the measurement data for the period between 7 March 2021 and 11 April 2021 are presented, since the greatest changes in the indoor climate and the MC of the beech glulam were measured during this period. At the beginning of the monitoring experiment, the recorded MC values were normalised in order to compensate measurement-related fluctuations (measurement-related offset). In addition, the measured MC values were T-compensated. The measured MC of the beech glulam was compared with the value of the respective wood equilibrium moisture content (EMC$_{calc}$), and this was calculated according to the Hailwood–Horrobin formula [15]:

$$\mathrm{EMC_{calc}} = \frac{1800}{W}\left(\frac{k\,h}{1\,k\,h} + \frac{k_1\,k\,h + 2\,k_1\,k_2\,k^2\,h2}{1 + k_1\,k\,h + k_1\,k_2\,k^2 h2}\right)$$

$$W = 330 + 0.452\,T + 0.00415\,T^2$$

$$k = 0.791 + 4.63 * 10^{-4}\,T - 8.44 * 10^{-7}\,T^2$$

$$k_1 = 6.34 + 7.75 * 10^{-4}\,T - 9.35 * 10^{-5}\,T^2$$

$$k_2 = 1.09 + 2.84 * 10^{-2}\,T - 9.04 * 10^{-5}\,T^2$$

$EMC_{calc}$: Equilibrium moisture content in %.
h: Relative humidity in decimal form in %/100.
T: Temperature in Fahrenheit.
W, k, $k_1$, $k_2$: Coefficients defined by the equations given above.
This served to show the extent to which the measured MC of the beech glulam approaches the $EMC_{calc}$ during the measurement period.

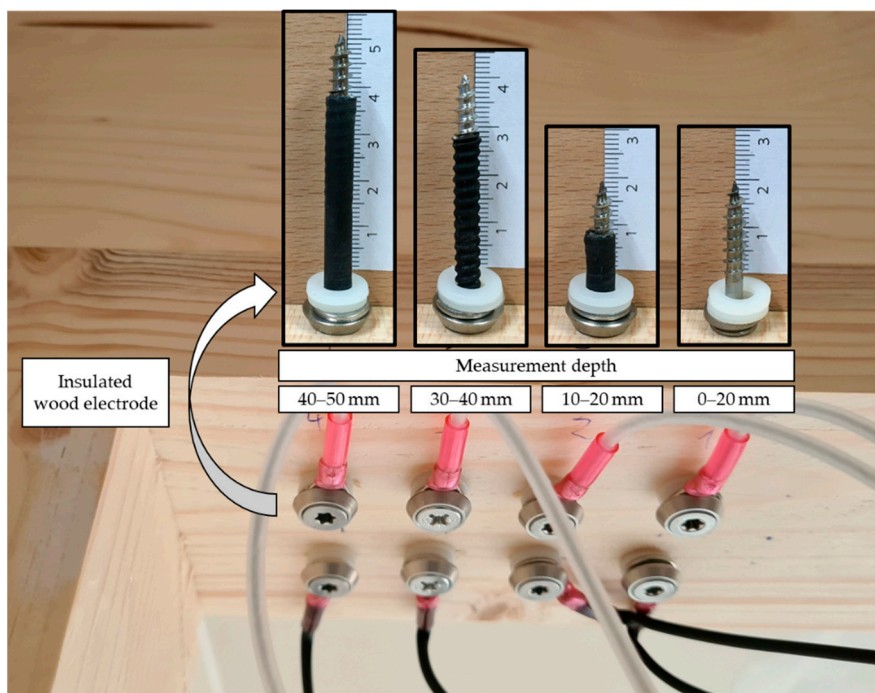

**Figure 2.** Resistance-based measurement of the wood moisture content of the beech glulam at different measuring depths.

### 2.3. Manufacture of Finger-Jointed Delamination Specimens and Testing under Varying Ambient Climates

The production of finger-jointed delamination specimens ($100 \times 175 \times 44$ mm$^3$) was carried out in accordance with EN 301 (2018) [4] and EN 14,080 (2013) [5] with an Ultra TT finger-jointing machine (Weinig Grecon GmbH & Co. KG, Alfeld, Germany) on a laboratory scale (Figure 3). A minifinger jointing cutter (Leitz GmbH & Co. KG, Oberkochen, Germany), with a finger length of 20–22 mm and a finger pitch of 6.2 mm, was used to cut the finger-jointing profiles. The cutting feed rate was set at 17 m min$^{-1}$.

The beech was conditioned based on the hygroscopic equilibrium of Loughborough and Keylwerth (in [13]) in a climate chamber at 20 °C over several weeks, until the specified bonding wood moisture contents ($MC_{bond}$) of $8 \pm 1\%$ (40% RH), $12 \pm 1\%$ (65% RH) and $15 \pm 1$ % (75% RH) were reached. Specimens in each group showed similar orientation of the annual rings. The adhesive systems were first applied in caterpillar shapes to the finger-joint profile (Figure 3) with defined application quantities (AQ) and mixing ratios (MR, resin: hardener) according to the manufacturer's instructions, and then applied to the finger-joints by hand with a brush. MUF (AQ 250 g m$^{-2}$, MR 100:50) and PUR (AQ 140 g m$^{-2}$) were applied on one side and PRF on both sides (AQ 190 g m$^{-2}$ each side, MR 100:20). The two-component systems, MUF and PRF, were applied after mixing resin and a commercially available hardener approved for beech wood bonding. The open assembly

time for bonding the specimens was 90 ($\pm$15) s and the closed assembly time was 30 ($\pm$15) s. Consistent manufacturing conditions were ensured by pressing the specimens individually. The pressing pressure was 11.5 N mm$^{-2}$ with a pressing time of 5 s. The adhesive bonding of the finger-joint profiles was tested for completeness using a UV lamp (Analytik Jena US, Uppland, USA) at 365 nm on the fluorescent PUR, in order to prevent application-related incorrect bonding. In total, 116 specimens were subjected to the delamination test according to EN 301 (2018) [4] within a period of 3–5 days after bonding, and 136 test specimens were subjected to the delamination test after three weeks of exposure alternating RH in a climate chamber (Memmert GmbH & Co.KG, Schwabach, Germany) (Figure 4).

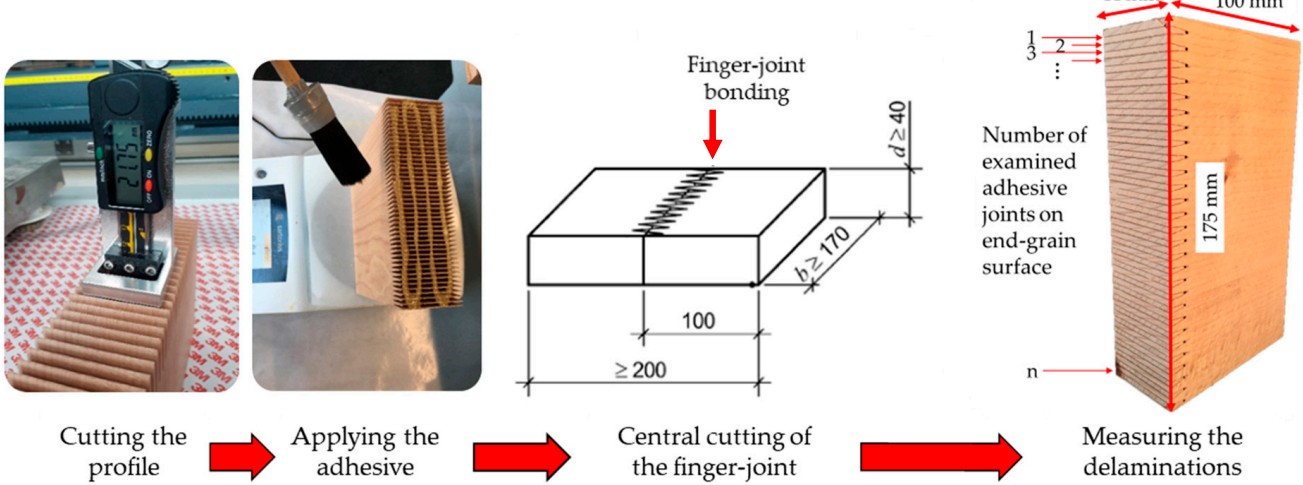

**Figure 3.** Manufacturing of delamination specimens and adhesive joints examined on the end-grain surface, technical drawing from [4] was adapted for this figure.

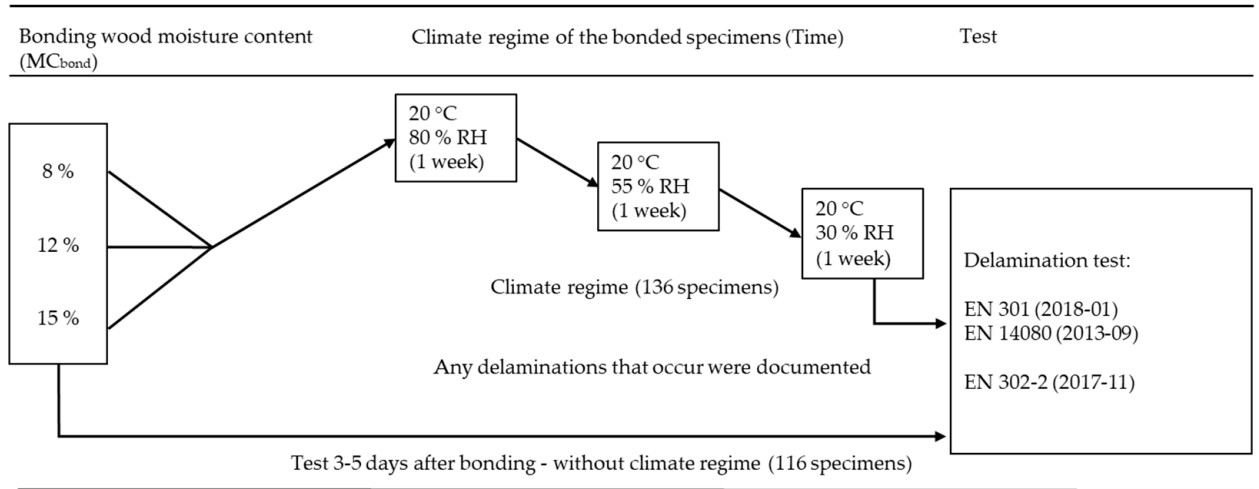

**Figure 4.** Course of the delamination test without and with previous alternating climate regime (RH = relative humidity).

The RH of the three-week alternating climate regime of the finger-jointed delamination specimens was oriented to the RH measured in the test building during the drying phase of the screed (between 23 March 2021 and 11 April 2021). It was reduced weekly, from 80% RH at the beginning to 55% RH and finally to 30% RH. In contrast to the test building of the beech glulam, T was kept constant at 20 °C during the entire alternating climate regime. Delamination occurring during the alternating climate regime was documented (Figure 4). The delamination test was divided into a 6 h boiling cycle (+30 min heating phase and 1 h cooling phase) and a subsequent drying cycle in an EasyQ delamination unit (Kempf GmbH, Hettingen, Germany) at 65 °C, 7.5% RH and 3 m s$^{-1}$ air speed (based on EN 302–2

(2017) [16]) for 20 h (+1 h heating phase). The total delamination was calculated according to the following formula [1]:

$$\text{Delam}_{tot} = 100 \; \frac{l_{tot, \; delam}}{l_{tot, \; glue \; line}}$$

$\text{Delam}_{tot}$: Total delamination in %.

$l_{tot, \; delam}$: Total length of delamination at the adhesive joints visible on the end-grain surface in mm.

$l_{tot, \; glue \; line}$: Total length of adhesive joints on end grain in mm.

According to EN 301 (2018) [4], the requirement for bonding is that the average total delamination of the finger-joints must be less than 10% and individual values must be less than 15%.

In order to obtain a benchmark for the extent of delamination at the single joint level, the maximum delamination per test specimen was calculated as follows, based on EN 14080 (2013) [5]:

$$\text{Delam}_{max} = 100 \; \frac{l_{max, \; delam}}{2 * l_{glue \; line}}$$

$\text{Delam}_{max}$: Maximum delamination in %.

$l_{max, \; delam}$: The maximum delamination length of an adhesive joint in mm.

$l_{glue \; line}$: The length of an adhesive joint in mm.

In addition, the specimen thickness was measured weekly over all $\text{MC}_{bond}$ ($n \geq 7$) during the alternating climate regime and during the delamination test before and after boiling and after drying, and set in relation to the initial thickness.

## 3. Results and Discussion

### 3.1. Monitoring of Climatic Conditions and Changes in Wood Moisture Content

After the insertion of liquid anhydrite screed (11 March 2021), the RH in the test building rose from 43% to 80–85% within two days (Figure 5). The T fluctuated between 8 and 16 °C. The heating and drying phase started on 23 March 2021 and the T increased continuously for 10 days, and up to 40–44 °C was reached. This T range was maintained until the end of the drying phase (10 April 2021). The RH peaked at 91% shortly after the start of the heating phase and then decreased continuously to 32% within 18 days.

The course of the MC of the beech glulam during the monitoring experiment and at different measuring depths was recorded (Figure 6). In comparison, the $\text{EMC}_{calc}$ is shown, which was calculated based on the T and RH in the test building. The greatest changes in the MC of the beech glulam were measured at a measuring depth of 0–20 mm (MC = 9–14%). The MC course in the near-surface measuring range (0–20 mm), with a delay of about one day, followed the changes in RH caused by the application of anhydrite screed and the drying process in the test building (Figure 5). The decrease of the MC did not start before a constant high T level (28 March 2021) was reached in the test building. At measuring depths of 40–50 mm, small changes in MC (9–10%) were measurable over the short monitoring experiment. Only the measuring electrode MC_P1_30.40 mm, which was installed in the area close to the end grain, showed larger changes in MC (9–12%) even at greater measuring depths. The $\text{EMC}_{calc}$, according to Hailwood and Horrobin [15], was approx. 8% at the time the anhydrite screed was applied, and thus coincided well with the measured MC. Afterwards, the $\text{EMC}_{calc}$ increased significantly and reached its maximum, with $\text{EMC}_{calc}$ = 20.7% (27 March 2021). Only the MC measured at the low measuring depths (MC_P1/P2_0–20 mm) approached the course of the $\text{EMC}_{calc}$.

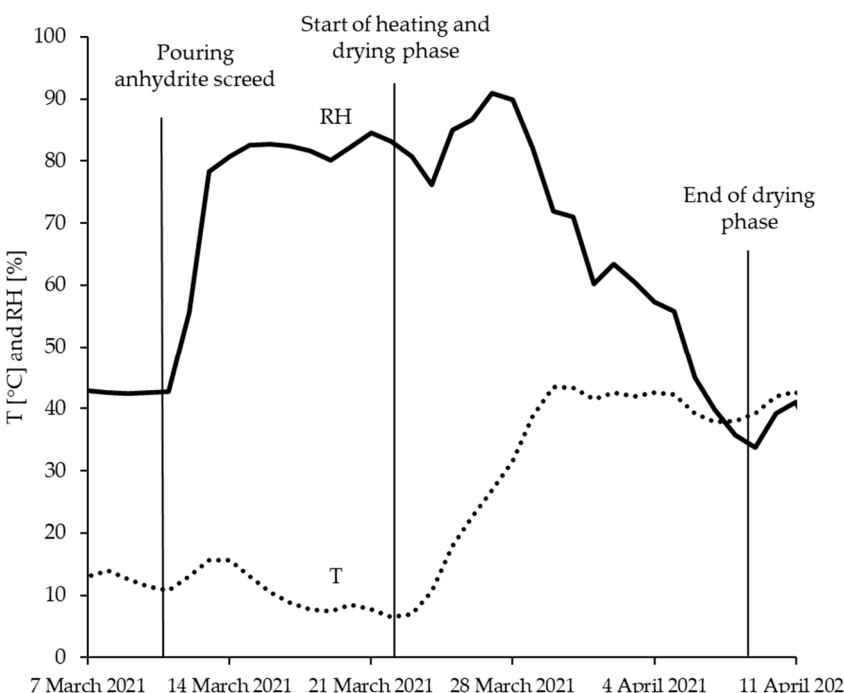

**Figure 5.** Course of temperature (T) and relative humidity (RH) in the test building during the monitoring experiment of the beech glulam.

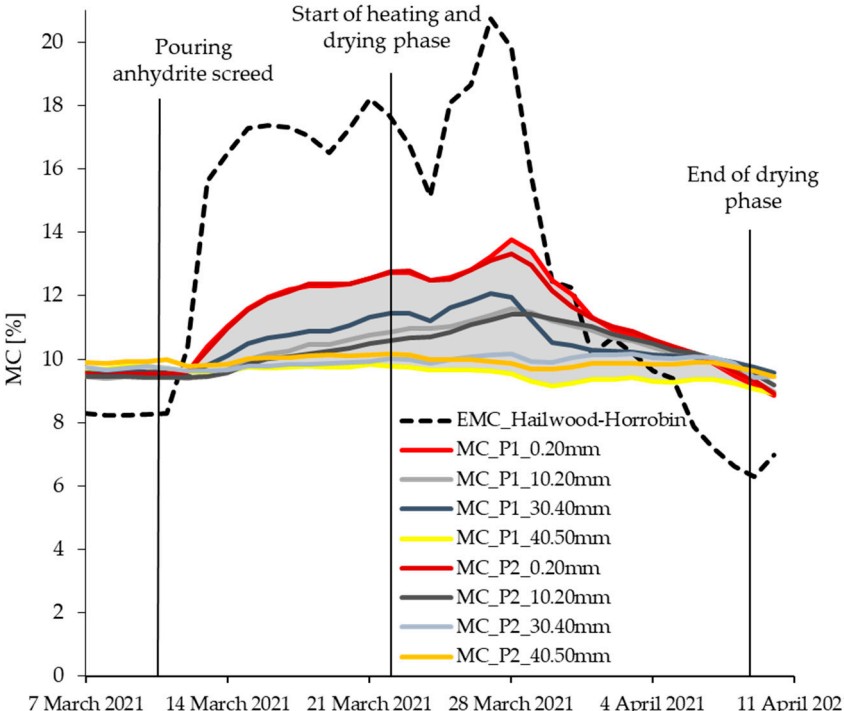

**Figure 6.** Wood moisture content (MC) of beech glulam at different measuring depths during monitoring experiment and calculated wood equilibrium moisture content (EMC$_{calc}$) resulting from the climatic conditions in the test building; measured values were temperature (T)-compensated and normalised.

The monitoring experiment in the test building showed extreme climate changes due to moisture released from the screed and a combination of ventilation intervals and heating. As expected, there was a great influence of the ambient climate, especially near the surface, and barely any changes were measurable inside the beech glulam. The resulting MC

gradients caused moisture-induced internal swelling-shrinkage stresses in the cross-section of the beech glulam and especially between the individual laminated layers [1,2,17–19], and led to sporadic delamination of the bonded surfaces in the beech glulam. The rapid drying of the beech glulam during the screed drying induced a stress reduction in the form of shrinkage cracks. The highest internal stresses occur in glulam when the MC differences within the glulam are at a maximum [2]. The maximum difference between the MC at the measuring points near the surface and those on the inside was measured at the end of the heating phase and when the room T reached approx. 40 °C. In order to reduce the stress on the glulam caused by the high humidity, a significantly shorter ventilation interval would be recommended, especially during the heating phase. Manual ventilation, which was carried out 1–3 times per day (weekdays) in this monitoring experiment, is only suitable to a limited extent, as high humidity prevailed at night and at weekends for a longer period of time and led to damage of the beech glulam. A possible solution to reduce the stresses in the beech glulam is a time-controlled ventilation system, whereby the influence of shorter ventilation intervals on the drying of the screed needs to be examined first, or a coating that reduces changes in moisture close to the surface. The greatest moisture stress on the finger-joint bonding occurs in the outer layers of the beech glulam, as the bonding is directly exposed to the climate in this area. This is important because the compression and especially the tension zones, which are located in the outer layers of the beech glulam, are particularly relevant for its load-bearing capacity and must be intact to ensure load-bearing safety [20]. In conclusion, the necessary delamination resistance of finger-joints must be assessed on the basis of the conditions of the outer glulam layers. The MC changes are dependent on T and RH [21], and EMC additionally depends on the duration of the climatic conditions [1]. In this monitoring experiment, MC changes resulted primarily from water vapor released from the screed. The T and ventilation in the monitoring experiment can be understood as a tool for controlling the RH. In a further section of this article, the focus of the alternating climate regime in the climate chamber was on the change in RH as a direct influencing variable, and the T was kept constant.

### 3.2. Delamination Resistance of Beech Finger-Joints in Varying Ambient Climates

The results of the delamination specimens tested as standard, without alternating climate regime (Figure 7a), and specifically for this test, with previous alternating climate regime (Figure 7b), were compared. To classify the individual values of the delamination specimens according to EN 301 (2018) [4] (requirement of the individual values of the delamination specimens <15%), the results are shown behind the adhesive systems.

The beech PRF bondings showed the lowest average delaminations in comparison with the other adhesive systems without (Figure 7a) and with (Figure 7b) previous alternating climate regime. The PRF met the requirements of EN 301 (2018) [4] in terms of mean delamination (<10%) and individual values (<15%) of the specimens. The $MC_{bond}$ had no influence on the resistance of the beech PRF bondings. The beech MUF bondings are the most delaminated in comparison. The requirements of EN 301 (2018) [4] were clearly not met by all beech-MUF combinations. The beech-PUR bonds showed a higher resistance at a higher $MC_{bond}$. At a $MC_{bond}$ of 15%, many of the specimens without and with alternating climate regime met the normative requirements. Across all tested combinations, no uniform trend was discernible with regard to the effect of alternating climate regime (Figure 7b). Hence, alternating climate regime was considered independently of the delamination test in Table 1. It summarises the average delamination characteristics of the beech finger-joints after completion of the three test procedures.

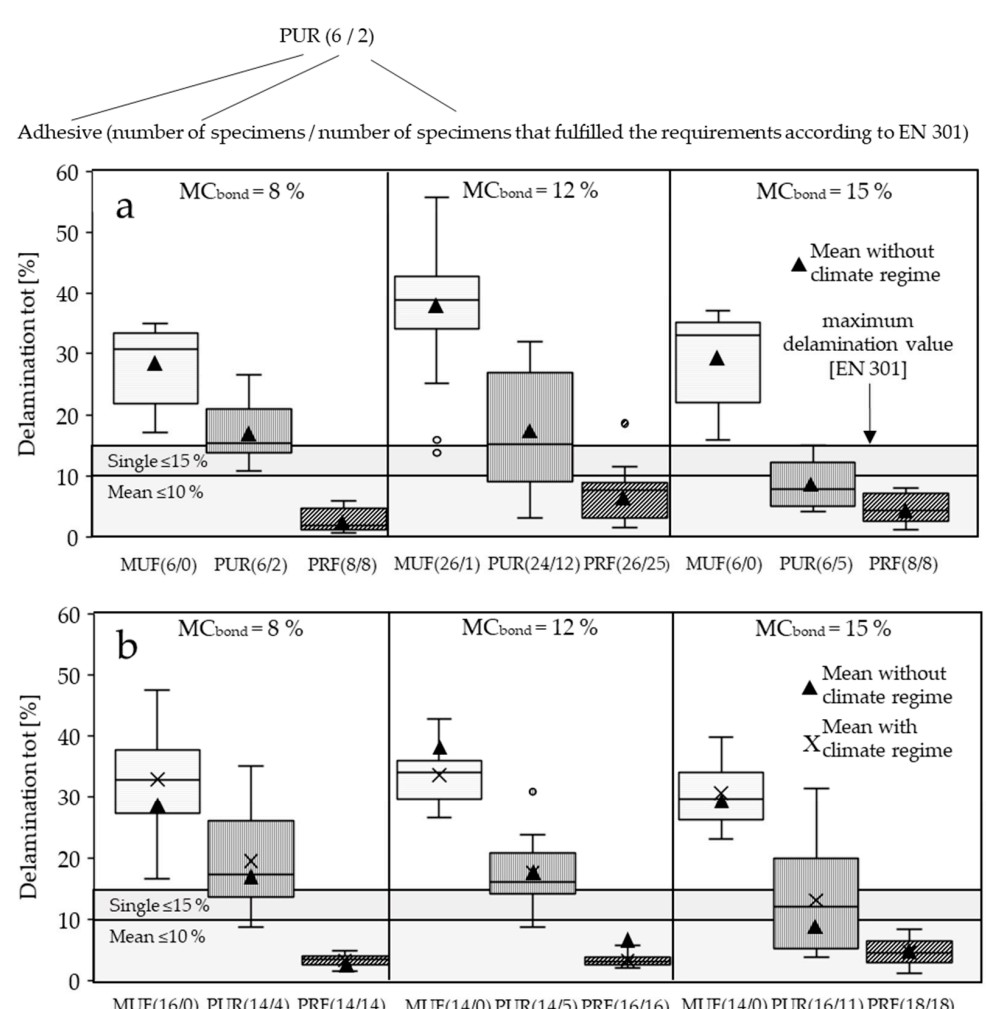

**Figure 7.** Delamination resistance of beech finger-joints depending on adhesive system and bonding wood moisture content (MC$_{bond}$) according to different test methods: (**a**) = Delamination test without alternating climate regime, (**b**) = Delamination test with previous alternating climate regime.

**Table 1.** Mean delamination characteristics of the beech finger-joints according to different test procedures.

| Adhesive | MC$_{bond}$ (%) | c | a | b | c | a | b |
|---|---|---|---|---|---|---|---|
| | | Mean Delam$_{tot}$ [%] | | | Mean Delam$_{max}$ [%] | | |
| MUF | 8 | 2.0 (±1.5) | 28.3 (±6.2) | 33.0 (±8.3) | 14.4 (±10.4) | 47.4 (±3.7) | 48.3 (±4.7) |
| | 12 | 1.9 (±2.0) | 37.8 (±9.1) | 33.7 (±4.8) | 14.6 (±9.4) | 49.3 (±2.3) | 47.8 (±5.4) |
| | 15 | 0.7 (±0.7) | 29.6 (±7.3) | 30.7 (±4.6) | 11.0 (±2.9) | 44.9 (±7.2) | 49.8 (±0.6) |
| PUR | 8 | 3.0 (±2.1) | 17.0 (±4.9) | 19.5 (±7.3) | 16.9 (±11.9) | 46.0 (±7.9) | 40.6 (±10.2) |
| | 12 | 1.6 (±1.6) | 17.3 (±9.2) | 17.6 (±5.3) | 18.1 (±16.8) | 39.9 (±12.1) | 42.6 (±7.9) |
| | 15 | 0.2 (±0.3) | 8.6 (±3.9) | 13.1 (±8.4) | 4.8 (±10.3) | 26.3 (±9.9) | 38.3 (±12.9) |
| PRF | 8 | 0.4 (±0.6) | 2.6 (±1.9) | 3.4 (±1.0) | 11.7 (±17.6) | 8.8 (±3.9) | 18.7 (±16.1) |
| | 12 | 0.6 (±0.7) | 6.7 (±3.7) | 3.3 (±1.0) | 13.1 (±16.6) | 25.7 (±13.6) | 20.7 (±15.1) |
| | 15 | 1.0 (±1.0) | 4.7 (±2.3) | 4.7 (±2.2) | 18.6 (±20.3) | 23.9 (±12.4) | 25.6 (±17.6) |

c = After 21 days at alternating climate regime, a = Delamination test without alternating climate regime, b = Delamination test with previous alternating climate regime, MC$_{bond}$ = bonding wood moisture content.

After three weeks of alternating climate regime (Table 1c) and prior to the actual delamination test, the average Delam$_{tot}$ across all MC$_{bond}$ was 1.5% for MUF, 1.6% for PUR and 0.7% for PRF. The lowest mean Delam$_{tot}$ after alternating climate regime, i.e., 0.2%, was documented for PUR bonded with 15% MC$_{bond}$. Already after the alternating climate (Table 1c), all adhesive systems showed partly complete delamination of single adhesive

joints. The high standard deviation shows that, at the same time, many of the specimens showed only small or no delamination after the alternating climate regime. After test methods a and b (Table 1), many specimens bonded with MUF showed complete delamination of single adhesive joints (complete if $Delam_{max} = 50$ %) and similarly, the PUR adhesives treated in the same way with 8% and 12% $MC_{bond}$ also showed this.

Compared to the delamination test cycle, the three-week alternating climate regime led only to low delamination percentages, which were within the standard requirements [4]. Due to the alternating climate, the differences in resistance of the adhesive systems were only visible to a limited extent. The alternating climate primarily depicted the short-term behaviour of beech finger-joint bonding under similarly extreme conditions, whereby all three tested adhesive systems showed approximately the same performance. The delamination test focuses more on the long-term durability of the bond and revealed clear differences between the adhesive systems (Figure 8).

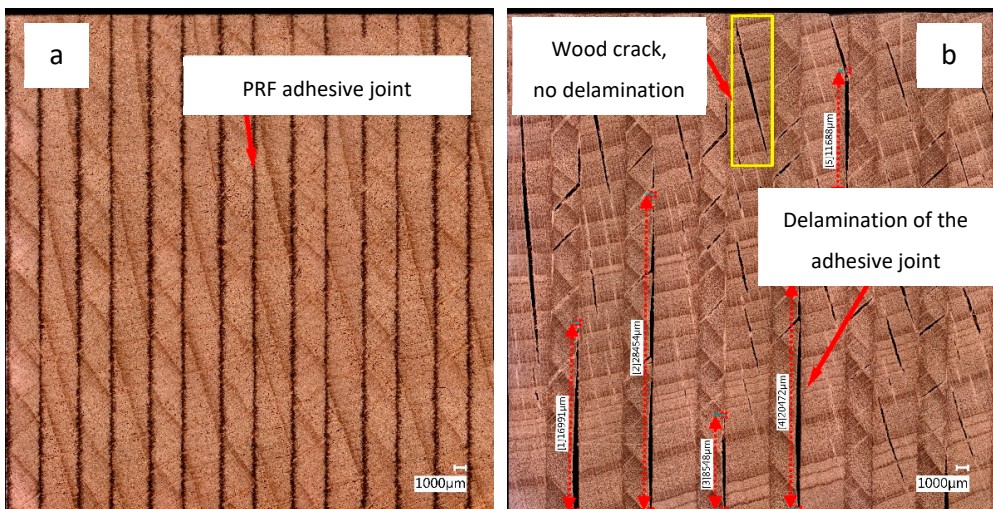

**Figure 8.** Delamination specimen before (**a**) and after (**b**) the delamination test according to [4], images taken with digital microscope VHX-5000 (Keyence, Osaka, Japan).

The alternating climate regime, which can occur, for example, when installing anhydrite screed, led to a pre-stressing of the bond, which could be seen in, among other things, the complete delamination of single adhesive joints. This can be associated with negative consequences for the long-term durability and load-bearing capacity of bonded beech building products, since MC changes can occur primarily in the outer layers and the delamination can thus occur with the greatest possible influence, especially in the compression and tension zones of the glulam. Consequently, further research is needed with respect to the influence of short- and long-term alternating climatic conditions on the strength of finger-joint bonding, since a climate-induced loss of strength was observed, as previously described [22,23]. Only the durability of the PUR bonding was increased by a higher $MC_{bond}$, which was also found for the strength of the PUR bonding in [24]. The other two adhesive systems reacted less to $MC_{bond}$. The use of a primer could possibly have further improved the wetting and bonding resistance of the PUR, which is currently being tested on various types of wood in [25]. Primers were not used in this study because there is currently no suitable application method for finger-joint bonding. An advantageous property of the PUR and PRF systems used is the higher elasticity compared to the more brittle MUF system. This could be the basis for the higher delamination resistance of the PUR and PRF compared to the MUF and can be explained by the possible effect of the PUR/PRF adhesive joint as an elastic "buffer element" between the swelling and shrinking finger-joint. In addition to the elasticity of the adhesive systems, their penetration into the wood structure may have an influence on the bonding performance [14,26,27] and possibly on the delamination resistance. Depending on the penetration and fixation of

the adhesives within the wood structure, different effects, such as overpenetration, anchoring and bulking effect and their influence on the dimensional stability of the wood and the delamination resistance of the bond, need to be investigated in more detail. The penetration of PRF components into the wood cell wall, observed by [27], is associated with accompanying wood modification (swelling reduction). This could be an important factor for reduced stress concentrations [28] and for high delamination resistance. This indication needs to be verified, as more information about the penetrated PRF components, whether water contained in the adhesive system or PRF adhesive components, is necessary to be able to make statements about higher adhesive-induced dimensional stability of the wood. A high delamination resistance of PRF has already been described several times in the literature [7,29] and also confirmed in this study. Under real conditions during short-term alternating climate, the PRF performed similarly to the other two systems. Under extreme stress during the delamination test, its advantageous properties such as high T and moisture resistance became apparent. A possible reason for the comparatively poorer performance of the MUF system could be the short closed assembly time before pressing the finger-joint. In various studies [30–34], a longer closed assembly time tended to lead to a higher bonding resistance of MUF adhesives, whereas PUR seemed to react less to the duration of the closed assembly time [32]. However, a longer closed assembly time, as known from surface bonding, is difficult to incorporate into the fast-cycle process of finger-joint bonding. The application of the MUF on both sides is probably advantageous for the resistance of the hardwood bonding, and should be preferred to the one-sided application in future investigations of this type. Due to the short processing time of the MUF, it was applied on one side in this laboratory test.

### 3.3. Effect of the Test Procedures on the Change in Thickness of the Delamination Specimens

The delamination specimens showed varying thickness changes during the alternating climate regime and the delamination test (Figure 9). Compared to the original specimen thickness after bonding, this increased by an average of 1.8% after 7 days of alternating climate regime and then decreased to 0.5% (after 14 days) and −0.6% (after 21 days). In the delamination test, greater changes in thickness were measured than in the alternating climate regime. After boiling, the delamination specimens showed an average thickness change of 5.2%, and after drying, 0.5%, compared to the original thickness.

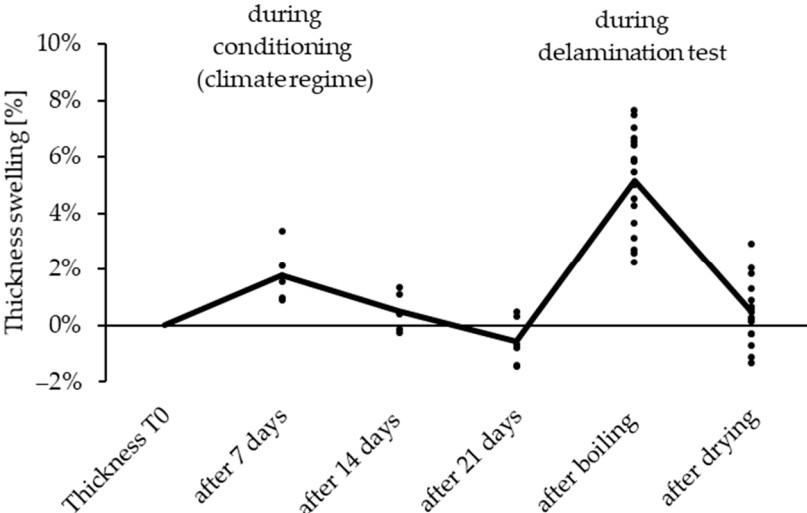

**Figure 9.** Change in specimen thickness during alternating climate regime and delamination testing.

The influence of the annual ring orientation on the change in thickness and on the delamination resistance was not considered, as it was difficult to clearly sort by annual ring orientation due to the normatively specified specimen dimensions [4]. Due to the swelling and shrinking anisotropy of wood [13], an influence of the annual ring orientation on the

resulting delamination percentages is expected. In particular, the fact that more than a quarter of the specimen surface is end-grain (Figure 3) results in a pronounced moisture transport during the test procedures. The dense beech wood, with its high moisture sorptivity, reacted with clear swelling and shrinking movements during the MC changes. The pronounced change in thickness during the delamination test shows the high stress on the bond and the relevance of an elastic adhesive joint. Above all, the MC changes caused in a short time during the delamination test significantly exceed the MC changes derived from the real installation situation in the monitoring experiment. The long-term resistance of the three adhesive systems under varying climatic conditions, which are possible in a real installation situation, remains to be tested. It should also be considered that the performance of the adhesive systems can be fundamentally different if processing parameters other than those used in this study are selected. In this study, we have tried to follow the recommendations of the adhesive manufacturers as closely as possible on a laboratory scale. Nevertheless, processing adjustments can lead to deviating results, which must be checked in each individual case.

## 4. Conclusions and Outlook

The main results of the monitoring experiment and the delamination test under varying ambient conditions were:

- In beech glulam, the monitoring experiment showed a clear MC gradient between the inner and near-surface wood. In particular, the outer layers of the glulam and the finger-joints located there are directly exposed to the moisture changes. In the outer layers, the beech wood reacted quickly to moisture changes in the ambient conditions;
- In the course of the three-week alternating climate regime, there are sometimes major delaminations of the finger-joint bonding. The adhesive systems used showed approximately the same delamination resistance after the alternating climate regime;
- PRF-bonded beech finger-joints fulfilled the requirements of EN 301 (2018) [4]. A high proportion of the specimens bonded with PUR were also able to pass the standard requirements at high $MC_{bond}$.

The following conclusions and outlook can be drawn from the results of this study:

- The load on wood adhesives in a practical construction application (at least initially) is different from that in a standardised performance test for the classification of adhesive systems (delamination test according to EN 301 (2018) [4]);
- The results show that delamination resistance remains a critical key property for the realisation of glulam made of beech wood;
- More elastic adhesive systems can have an advantage over more brittle systems in the case of moisture-related dimensional changes;
- The bonding performance of the adhesive systems is dependent on the bonding parameters, as determined here for PUR and $MC_{bond}$;
- The influence of a long-term alternating climate regime on the properties of finger-joint bonding is to be investigated;
- It is to be examined whether the annual ring orientation of the delamination specimens has an influence on the delamination of the bonding.

**Author Contributions:** Conceptualization, H.S. and M.S.; Data curation, S.K. and C.F.-Z.; Formal analysis, H.S. and M.S.; Funding acquisition, H.M.; Investigation, H.S., M.S., S.K. and C.F.-Z.; Methodology, H.S. and M.S.; Project administration, H.M.; Resources, H.M.; Supervision, C.B. and H.M.; Visualization, H.S.; Writing—original draft, H.S.; Writing—review & editing, C.B. and H.M. All authors have read and agreed to the published version of the manuscript.

**Funding:** This research was funded by the German Federal Ministry for Economic Affairs and Energy (BMWi) through the Central Innovation Programme for small and medium-sized enterprises (SMEs) grant number FKZ 16KN042025. The authors acknowledge support by the Open Access Publication Funds of the University of Goettingen.

**Acknowledgments:** The authors would like to express their sincere thanks to Weinig Grecon GmbH & Co. KG (Alfeld, Germany) for providing the finger-jointing line and expertise, to Bau-Fritz GmbH & Co. KG (Erkheim, Germany) for making the test building available and for their support during the study, to NOKA Holzverarbeitungs-GmbH (Saterland, Germany) for producing and providing the beech glulam and to the three adhesive manufacturers for providing the adhesive systems. We would also like to thank Philipp Schlotzhauer, Dieter Varel, Bernd Bringemeier and Gerhard Birke for initiating and supporting the project.

**Conflicts of Interest:** The authors declare no conflict of interest.

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
