# Peer review of "Monitoring of Beech Glued Laminated Timber and Delamination Resistance of Beech Finger-Joints in Varying Ambient Climates"

_forests, doi:10.3390/f12121672_

Round 1

Reviewer 1 Report

The article „Monitoring of beech glued laminated timber and delamination resistance of beech finger-joints in varying ambiet climates” concerns very important topic from the perspective of beech wood structural applications. Although the subject is not really new, Authors presented the results which constitute a valuable systematization and supplementation of the existing knowledge about such popular joints system. The varying climate conditions and their effect on structural materials are the objective of many ongoing research but due to the strong influence of both wood-related and resin-related properties is still not fully understood and described. This paper is especially valuable since it refers to various adhesive systems (PUR, MUF, PRF), various conditions and testing methods (laboratory and building simulation).

Abstract:

The abstract is well written and provides the information about the most important parts of the study. It introduces the Reader to the subject, presents what the conducted study is aimed for, summarizes the methodology and the most important conclusions.

The introduction:

This part is written in a very interesting way. Before proceeding to the experimental part of the article, it provides all the necessary information on each aspect of the conducted research. The literature is selected in a proper way. This part ends with the clear statement what the aim of the study is which is clear and justified from the Reader perspective.

Line 53: Since the research is partially focused on the adhesive systems maybe it would be beneficial to discuss briefly which parameters exactly or what this parameters are related to.

Line 55: Can you provide the examples of the adhesive systems in which case the requirements were not met?

Methodology and Results:

In general this part is presented clearly and the applied methods were selected in a proper way, in accordance with appropriate standards. In case of results and discussion in my opinion it is very detailed description, each observation was explained in a scientific way. Since there were that many studies on finger joints and variable climate changes in case of structural applications it would be interesting to compare these observations with the conclusions from other studies (especially when it comes to the various adhesive formulations etc.) (this is just a general suggestion).

Line 85-88: What was the selection criterion for selecting these 3 binding agents?

Line 88: It would be nice to mention the name of the adhesive producers.

Line 147: Does experimental design assume some any specific quantity of the glue? Is it different for different adhesive systems?

Line 150: What kind of hardener? Commercial one? Maybe Authors should consider to describe the adhesives formulations in more detailed way.

Table 1: Authors may consider to add some basic statistics such as homogeneous groups.

Conclusions:

Conclusions and Outlook are specific and correct, moreover, they correspond with the aim presented in the last paragraph of introduction.

In summary, in my opinion this is a well written paper which will be valuable reference for scientists interested in the structural materials and their behavior in the changing climate conditions. The text is also well prepared taking into account linguistic and editorial correctness.

Author Response

Dear Reviewer,
thank you very much for reviewing our article.
You will find our comments on the additions in the appendix.

Yours sincerely,
Hannes Stolze

Reviewer 2 Report

Point 1: "On the one hand" (Line 50) and "On the other hand" (Line 53) are not commonly used in the scientific papers.

Point 2: Though the details of beech finger-joints are explained in the article, adding a technical drawing about the details of beech finger-joints to Figure 3 to explain it more clearly to the readers.

Point 3: Please specify the application of MUF, PUR, and PRF. (Line 149)

Point 4: Figure 5 and Figure 6 lack the information about the date of "end of drying phase".

Point 5: There is no phenomenon of the beech finger-joints delamination. If it is possible, I suggest to provide some representative microscopic images (SEM micrographs) of the beech finger-joints.

Author Response

(The authors gave the same response as above.)
